# Characterization of the Isocitrate Dehydrogenase Gene Family and Their Response to Drought Stress in Maize

**DOI:** 10.3390/plants12193466

**Published:** 2023-10-02

**Authors:** Ningning Wei, Ziran Zhang, Haoxiang Yang, Die Hu, Ying Wu, Jiquan Xue, Dongwei Guo, Shutu Xu

**Affiliations:** 1Key Laboratory of Biology and Genetic Improvement of Maize in Arid Area of Northwest Region, Ministry of Agriculture and Rural Affairs, College of Agronomy, Northwest A&F University, Yangling 712100, China; weiningnwafu@163.com (N.W.); zhangziran@nwafu.edu.cn (Z.Z.); 2021050058@nwafu.edu.cn (H.Y.); 15922359154@163.com (D.H.); wuying313@nwafu.edu.cn (Y.W.); xjq2934@163.com (J.X.); 2Maize Engineering Technology Research Centre, Yangling 712100, China

**Keywords:** maize, isocitrate dehydrogenase (IDH), drought, association analysis

## Abstract

Isocitrate dehydrogenase (IDH) is a key rate-limiting enzyme in the tricarboxylic acid cycle and acts in glutamine synthesis. IDH also participates in plant growth and development and in response to abiotic stresses. We identified 11 maize *IDH* genes (*ZmIDH*) and classified these genes into *ZmNAD-IDH* and *ZmNADP-IDH* groups based on their different coenzymes (NAD^+^ or NADP^+^). The *ZmNAD-IDH* group was further divided into two subgroups according to their catalytic and non-catalytic subunits, as in *Arabidopsis*. The *ZmIDHs* significantly differed in physicochemical properties, gene structure, conserved motifs, and protein tertiary structure. Promoter prediction analysis revealed that the promoters of these *ZmIDHs* contain cis-acting elements associated with light response, abscisic acid, phytohormones, and abiotic stresses. *ZmIDH* is predicted to interact with proteins involved in development and stress resistance. Expression analysis of public data revealed that most *ZmIDHs* are specifically expressed in anthers. Different types of *ZmIDHs* responded to abiotic stresses with different expression patterns, but all exhibited responses to abiotic stresses to some extent. In addition, analysis of the public sequence from transcription data in an association panel suggested that natural variation in *ZmIDH1.4* will be associated with drought tolerance in maize. These results suggested that *ZmIDHs* respond differently and/or redundantly to abiotic stresses during plant growth and development, and this analysis provides a foundation to understand how *ZmIDHs* respond to drought stress in maize.

## 1. Introduction

Isocitrate dehydrogenase (IDH), a metabolic enzyme that converts oxidized NAD (NADP) to NADH (NADPH) and catalyzes the oxidative decarboxylation of isocitrate to α-ketoglutarate, is regarded as a key rate-limiting enzyme in the tricarboxylic acid cycle [1,2]. The tricarboxylic acid cycle is a crucial pathway not only for sugar metabolism but also for the ultimate oxidation of lipids and proteins and nucleic acid metabolism to carbon dioxide and water [3]. In addition, IDH participates in the biosynthesis of α-ketoglutarate, which can promote the synthesis of glutamate and other related amino acids [4]. Consequently, the activity of IDH is considered essential to the life-sustaining metabolism of plants.

In *E. coli*, EcIDH is a NADP^+^-dependent enzyme with a 7000-fold preference for NADP^+^ over NAD^+^ [5,6]. In *Saccharomyces cerevisiae*, the protein IDH1 contains the non-catalytic subunit, which mainly exercises regulatory functions, and IDH2 contains the catalytic subunit [7,8]. NtIDHb and NtIDHc from tobacco can replace the function of regulatory subunit IDH1 in *idh* mutant yeast, and heteromeric enzymes consisting of IDHa and IDHb or IDHa and IDHc can complement the catalytic function of *idh2* mutants [9]. Proteins that encode distinct IDH subunits have been successfully isolated from humans [10], monkeys [11], pigs [12], and bovines [13], and their functional properties have been investigated. However, there was limited research about the IDH genes in plants, particularly in maize.

In model plant *Arabidopsis*, IDHs can be separated into two groups: NAD^+^-dependent IDH (NAD-IDH) and NADP-dependent IDH (NADP^+^-IDH) based on the characteristics of their required coenzymes (NAD^+^ or NADP^+^) [14]. In plant cells, the proteins of NAD-IDHs are mainly distributed in mitochondria, but the NADP-IDHs are mainly distributed in the cytoplasm, chloroplasts, plastids, mitochondria, and peroxisomes [15,16]. The NADP-IDH-catalyzed production of α-ketoglutarate provides a carbon skeleton for ammonia uptake and assimilation by plant cells, and NADPH can maintain intracellular redox equilibrium and help plants resist oxidative stress [17]. The discovery of an interaction between two *NADP-IDH* genes (*Os01g0654500* and *Os05g0573200*) and *GH3* (Gretchen Hagen 3) in rice suggested the interactive crosstalk between hormone transduction pathways and glutathione metabolic pathways and co-regulation of seed germination [18]. The response of *NADP-IDH* genes to various abiotic stresses has been observed in a variety of plants [19,20]. In sugarcane, the expression level of *SoNADP-IDH* (GenBank accession number: KF808326) changed in response to pathogenic bacterial infestation and diversity abiotic stresses [21]. In pea, the expression of the *NADP-IDH* gene (GenBank accession number: AY509880) will increase under water stress [22]. In wheat, the expression of the *NADP-IDH* gene (GenBank accession number: AK353917.1) was down-regulated under chronic and transient nitrogen stress, with the lowest expression at six hours of transient nitrogen stress [23]. In maize, an *NADP-IDH* gene named *ZmIDH2* (*Zm00001d011487*) was cloned and shown to be strongly expressed in roots and young embryos, with higher expression and enzyme activity when subjected to salt and drought stresses [24]. Meanwhile, overexpressing the maize *NADP-IDH* gene (GenBank accession number: ACF88442) in *Arabidopsis thaliana* significantly increased the salt tolerance of transgenic plants [25]. These findings suggested that *NADP-IDH* genes can regulate responses to abiotic or biotic stress.

However, most previous studies focused on gene expression level and its variation under stress, with little attention to understanding function or identifying favorite alleles of *NAD-IDH* in plants. The model plant *Arabidopsis* contains six potential *NAD-IDH* genes. The non-catalytic subunit of the *idh-II* mutant exhibits significantly lower in vitro enzyme activity than that of the wild type, but growth and development are unaffected, suggesting that the missing *idh-II* might be replaced by other *NAD-IDH* [26]. Studies in tobacco NAD-IDH have shown that the catalytically active subunit NtIDHa cooperates with one or more of the non-catalytic subunits NtIDHb or NtIDHc to function [9]. Overexpressing the *Zm00001d008244* gene (*ZmIDH*) from maize in *Arabidopsis thaliana* will decrease fertility compared with the wild type [27]. Overall, the fluctuations in IDH activity in maize will affect plant growth and development.

As one of the main limiting factors in maize growth and development, drought forces plants to produce more reactive oxygen species (ROS) and increases free ammonium in cells, which is part of the metabolic pathway for glutamine synthase. In this pathway, *ZmIDHs* provide the necessary α-ketoglutarate and lower the toxicity of plants [28]. Thus, *ZmIDHs* were considered to work on plant drought response, with different potential regulation by different *ZmIDHs*. In this work, we used a comparative genomics and bioinformatics approach to identify and characterize *IDH* family genes in maize. Additionally, we tried to mine the favorite alleles of *ZmIDH* genes to confirm the function of these genes in response to drought [29]. The findings will provide new insights into the genetic effects and molecular functions of the *IDH* gene family in maize.

## 2. Results

### 2.1. Identification of ZmIDH in Maize

By using BioMart and BLASTP tools with the conserved structural domains of isocitrate dehydrogenase in the Pfam database of maize, we identified a total of 15 related genes, a number that is consistent with previous work in maize [27]. After the removal of the repeated genes and those not included in the most recent B73 reference genome (version 5.0), 11 *ZmIDH* genes remained and were used for further investigation (Table 1). Based on their different coenzyme dependences (NAD^+^ or NADP^+^) and their locations on the chromosome, these genes were named *ZmIDH1.1*–*ZmIDH1.6* and *ZmIDH2.1*–*ZmIDH2.5*, where *ZmIDH1.1*–*ZmIDH1.6* represent the six NAD^+^-dependent IDH genes (*ZmNAD-IDHs*) and *ZmIDH2.1*–*ZmIDH2.5* represent the five NADP^+^-dependent IDH genes (*ZmNADP-IDHs*). These genes are distributed on nine chromosomes, with none on chromosome 7. The lengths of the encoded proteins of *ZmIDHs* ranged from 211 to 503 amino acids (aa), with an average of 364 aa. All ZmNADP-IDHs, except ZmIDH2.5, are longer in protein length and have a higher molecular weight and hydrophilicity than those in NAD-IDHs. All ZmNADP-IDHs are predicted to be located in the cytoplasmic fraction, while all NAD-IDHs are predicted to be located in the mitochondrion fraction, consistent with the idea that the various coenzymes might have different molecular functions and act in diverse biological activities [15,16].

To investigate the evolution of IDHs in plants, protein sequences of 10 AtIDHs from *Arabidopsis thaliana*, seven OsIDHs from *Oryza sativa*, and nine SbIDHs from *Sorghum bicolor* were collected. Furthermore, they were used to construct the phylogenetic tree and split into the two subfamilies based on their coenzymes (NAD^+^ and NADP^+^) according to the classification in Arabidopsis (Figure 1). Furthermore, the NAD-IDHs were further divided into two clusters based on their catalytic activity. Previous studies showed that the genes AtIDH4 (At5g03290) and AtIDH8 (At3g09805) in Arabidopsis encoded proteins with catalytic subunits [30]. Based on this, ZmIDH1.1, ZmIDH1.2, and ZmIDH1.5, which clustered with AtIDH4 and AtIDH8, may be catalytically active, and the ones that did not cluster with these two proteins, ZmIDH1.3, ZmIDH1.4, and ZmIDH1.6, were considered to lack catalytic activity. These findings imply that NAD-IDH and NADP-IDH proteins may have different structures and functions and that different types of IDH proteins cannot be replaced.

### 2.2. Gene and Protein Structure Analysis of ZmIDHs

To get further insight into the characterization of the ZmIDH family, ten conserved motifs were identified among ZmIDHs using the MEME website (Figure 2B). Among them, motif 5 was found in all ZmIDH proteins, and motifs 1, 2, 4, 6, 7, and 8 were found in most IDH proteins, suggesting that these motifs are conserved in ZmIDHs. Motif 3 was only detected in NAD-IDH proteins, and motifs 9 and 10 were only identified in NADP-IDH proteins, suggesting they were formed during evolution and that these different motifs underlie functional differentiation. Furthermore, the tertiary structures of the IDH proteins were predicted to be predominantly comprised of α-helices, β-folds, and random coils (Appendix A), with marked differences between NAD-IDH and NADP-IDH and between catalytically active NAD-IDH and non-catalytically active NAD-IDH. Overall, these results consist of a phylogenetic tree and the specialized functions of different ZmIDH types.

Furthermore, *ZmIDH* gene structures were predicted, and they showed a very different distribution of exons and introns in the two different subfamilies of *ZmIDH* genes, but similar for genes in the same subgroup (Figure 2C). The results of gene structure and conserved motif predictions are consistent with the phylogenetic tree. High similarity in both conserved motifs and gene structures was observed in three gene pairs: *ZmIDH1.2* and *ZmIDH1.5*, *ZmIDH1.4* and *ZmIDH1.6*, and *ZmIDH2.2* and *ZmIDH2.4*. Notably, *NAD-IDH* genes exhibited substantial differences in intron regions, implying potential functional divergence. Moreover, *NADP-IDH* genes displayed more exons with fewer nucleotides compared with *NAD-IDH* genes. These disparities in exon numbers and lengths explain the variations in coding sequence lengths, ranging from 211 bp to 503 bp, among these genes (Table 1).

### 2.3. Colinearity and Duplication of ZmIDH Family Genes

Given that maize is a paleo-tetraploid plant, we conducted gene duplication analysis for *ZmIDH* genes. Gene duplication, including tandem or segmental duplication, is commonly considered a critical mechanism associated with gene family expansion and complexity [31]. Previous studies have defined tandem duplication events as chromosome regions within 200kb containing two or more genes [32]. For the 11 *ZmIDHs*, the collinearity analysis indicated the presence of gene duplications between three gene pairs: *ZmIDH1.2* and *ZmIDH1.5*, *ZmIDH1.3* and *ZmIDH1.4* as well as *ZmIDH2.2* and *ZmIDH2.4*. In contrast, these three gene pairs are located on different chromosomes, identified as segmental duplication events (Figure 3A), and exhibit highly homologous sequences (Figure 2 and Appendix A). Meanwhile, we calculated the ka (the ratio of the number of synonymous substitutions per synonymous sites) and ks (the ratio of the number of non-synonymous substitutions per non-synonymous sites), and the ka/ks ratios of these three gene pairs were less than 1 (Appendix A), indicating that the *ZmIDH* genes may have been affected by negative selection during maize evolution.

The number of IDH genes in maize (11) is greater than that in rice (7) and sorghum (9); the shared evolutionary relationships of these IDHs were deciphered through comparative co-linearity analysis among different species (Figure 3B). The results suggested that *ZmIDH1.2* and *ZmIDH1.5* are orthologous to the rice gene *LOC_Os01g16900* and the sorghum gene *Sb03g011050*, while *ZmIDH1.4* is orthologous to the rice gene *LOC Os02g38200* and the sorghum gene *Sb04g024840*. These genes are annotated as IDH, NAD^+^-dependent, and are involved in the TCA cycle and L-glutamine biosynthesis III. This suggests that *ZmIDH1.2*, *ZmIDH1.5,* and *ZmIDH1.4* may retain similar functions as their orthologs. Interestingly, the rice gene *LOC_Os01g46610* shares a homology with the maize genes *ZmIDH2.2*, *ZmIDH2.3*, and *ZmIDH2.4*, while the sorghum gene *Sb003G241500* is homologous to two paralogous genes, *ZmIDH2.2* and *ZmIDH2.4*. *ZmIDH2.3* likely underwent further functional evolution in maize. Both rice and sorghum genes are annotated as NADP-dependent IDHs, exclusively participating in L-glutamine biosynthesis III but not in the TCA cycle. Thus, it suggested that these genes evolved into different functions of the *NAD-IDH* and *NADP-IDH* genes in plants.

### 2.4. Analysis of Promoter cis-Elements and the Protein Network of ZmIDH Genes

To clarify the regulation of expression, the 2000 bp sequences upstream of the promotor of the *ZmIDHs* were selected to predict the cis-acting elements using the PlantCARE database. Multiple cis-elements, including those responding to environmental stresses, growth hormones, and endogenous signals related to plant growth and development, were detected (Figure 4A). The major promoter elements are light-responsive elements, which is consistent with the function of these genes in the TCA cycle in photorespiration (Figure 4B). Elements including response to jasmonic acid, auxin, defense, and stress were detected in the majority of *ZmIDH* genes (Figure 4C). Other elements were only detected in a few *ZmIDHs*. For example, defense and stress response elements were detected only in *ZmIDH1.3* and *ZmIDH1.6*, cell cycle regulation elements were detected only in *ZmIDH1.2* and *ZmIDH1.4*, and the MYB binding site of a drought-inducibility element was detected only in *ZmNADP-IDHs* and *ZmIDH1.3.* These findings suggested that *ZmIDHs* may vary in function based on their different coenzymes.

To further investigate the regulation network of the ZmIDH proteins, an interactive network was constructed using the 11 ZmIDH proteins. In the network, the interaction between NADP-IDH proteins and NAD-IDH proteins was detected, as was the interaction between NAD-IDHs with the catalytic subunit and non-catalytic subunit, consistent with the previous finding in tobacco that activity of the catalytic subunit requires binding to one or more non-catalytic subunits [9]. However, protein interaction did not occur within the three respective classes of NADP-IDH proteins: catalytic NAD-IDH and non-NAD-IDH (Figure 5A).

Next, a wider network was constructed to predict the interactions between ZmIDHs and other proteins. Furthermore, a total of 24 proteins were predicted to interact with both NAD-IDH and NADP-IDH, which included Succinyl-CoA synthetase subunit alpha (*GRMZM2G039251*, *GRMZM2G072054*), citrate synthase 1 (*GRMZM2G063851*), citrate synthase 2 (*GRMZM2G064023*), malate dehydrogenase 7 (*GRMZM2G068455*), malate dehydrogenase 12 (*GRMZM2G072744*), and 2-ketoglutarate dehydrogenase 1 involved in glutamine synthesis (*GRMZM2G142863*) (Figure 5B). The GO enrichment results indicated that proteins interacting with ZmIDHs are mainly involved in biological processes such as cellular respiration, aerobic respiration, the tricarboxylic acid cycle, and the citrate metabolic process (Figure 5C and Appendix A). The KEGG enrichment results reveal that the enriched pathways of the interacting proteins are primarily concentrated in the citrate cycle (TCA cycle) and related pathways (Figure 5D). These processes are closely associated with mitochondrial respiration, and drought stress increases the demand for respiratory ATP to support cellular metabolism [33]. The functional analysis of proteins interacting with ZmIDHs suggests that ZmIDHs proteins might be involved in the response to drought stress.

### 2.5. Expression Patterns of ZmIDHs

Gene expression can exhibit different patterns during maize growth and development or in response to various environmental conditions, and understanding the spatiotemporal patterns can provide insight into gene function. Here, expression levels of *ZmIDHs* were examined in nine tissues from two different inbred lines (the reference genome B73 and the elite inbred KA105 selected in our lab) by real-time quantitative PCR. Additionally, the expression patterns of *ZmIDHs* were analyzed using the RNA-seq data from 79 tissues from B73 [34]. The results of real-time quantitative PCR showed specific expression of most *ZmIDH* family members in the anthers, especially *NAD-IDHs* (Figure 6A). This phenomenon also existed in the RNA-seq data (Figure 6B). The clustering results showed the separation of *NAD-IDH* and *NADP-IDH* genes according to their tissue expression patterns, further indicating that different types of *ZmIDH* genes have different functions in different maize tissues (Figure 6).

Next, the expression level of *ZmIDHs* under abiotic stress at different developmental stages was also analyzed using two RNA-seq data sets (Figure 7). By using our RNA seq data at the seedling stage under normal conditions and under abiotic stresses, including PEG and ABA stress, significant differential expression was identified in both leaves and roots (Figure 7A). Overall, there was higher expression of *ZmIDH1.3*, *ZmIDH1.4*, and *ZmIDH1.6* with non-catalytic subunits and the *NADP-IDH* gene *ZmIDH2.1* in leaves than roots, and expression level increased with increased stress duration in leaves. In contrast, other *ZmIDHs* showed higher expression in roots than leaves and decreased expression with longer stress times. In another published RNA-seq data under drought stress at different stages [35], all *ZmIDHs* were highly expressed in tassels under drought treatment, especially at the V16 and R1 stages. *NADP-IDHs* showed a significant increase in expression under drought stress, while *NAD-IDH* showed negligible changes in expression (Figure 7B). The real-time qPCR data showed all these genes would respond to multiple abiotic stresses, including 200 mM NaCl, low nitrogen, and 15% PEG 6000 stress (Appendix A). Furthermore, *ZmIDH1.4* in *NAD-IDH* and *ZmIDH2.2* in *NADP-IDH* were more responsive to drought stress in both leaves and roots, suggesting that these genes may regulate multiple abiotic stresses at different developmental stages.

### 2.6. Candidate Gene Association Analysis of ZmIDHs Gene to Seed Suirvival Rate under Drought

To determine whether *ZmIDHs* respond to drought stress, candidate gene association analysis was conducted based on the previously reported seed survival rate (SSR) phenotype and genotype of 368 maize inbred lines [29]. By extracting the SNPs located 2 kb upstream of the start codon or the gene region of the 11 *ZmIDHs*, 389 SNPs were identified, and 28 SNPs were associated with SSR using a mixed linear model at *p* < 0.05 (Table 2). Of these, chr5_180550035 and chr5_180550432 from *ZmIDH1.4* exhibited the highest effect values (*p* < 0.001 and R^2^ = 6.706) (Table 2).

Among them, four significant SNPs (chr5_180550035, chr5_180550432, chr5_180550695, and chr5_180550539) in *ZmIDH1.4* associated with SSR under seedling drought stress were used for haplotype analysis based on the 368 maize germplasms. With complete linkage between chr5_180550035 (*p* = 2.09 × 10^−6^) and chr5_180550432 (*p* = 2.09 × 10^−6^), the chr5_180550035 with a higher significant level was used for haplotype analysis. The SSR of Hap1 (GGC) was significantly higher than that in the other two haplotypes (Figure 8C), confirming that *ZmIDHs* act in response to drought regulation.

## 3. Discussion

As one of the three major crops in the world, maize has significant economic and nutritional value. One of the main limiting factors for maize production is drought, which causes a high decrease in grain yield. Drought causes the buildup of reactive oxygen species (ROS) that damage plants and the extent of ROS damage is influenced by the NADPH/NADP ratio in chloroplasts and mitochondria and the activity of NAD(P)H oxidase in cell membranes [36]. IDH is an important rate-limiting enzyme that catalyzes NAD(P)H production in the tricarboxylic acid cycle, which is closely related to respiration and metabolic activities in plants. To understand the function of the IDH genes and investigate if these genes act in response to drought resistance in maize, 11 *ZmIDH* family members were identified and characterized.

According to the conservative motif prediction, gene structure, and phylogenetic tree analysis (Table 1, Figure 1), *ZmIDHs* could be divided into two subfamilies (*ZmNAD-IDH* and *ZmNADP-IDH*) based on their different coenzymes, similar to that in *Arabidopsis* [37]. Meanwhile, phylogenetic analysis, where the two different types of *ZmIDH* from four species (maize, rice, sorghum, and arabidopsis) formed two different clusters, revealed the differentiation of *ZmIDHs* before the evolution of dicots and monocots, and the different number of family members in different species reflects different gene duplication and expansion in each species [38,39]. Moreover, the ka/ks ratios of the three gene duplication pairs identified in this study were less than 1, indicating that these genes were subject to purification selection during evolution; this might be an important role of IDH in plant growth and development. In addition, the expression patterns of *ZmIDHs* under abiotic stresses showed that *ZmIDH1.4* and *ZmIDH2.2* were more responsive to drought stress in leaf and root (Appendix A). Finally, the results of promoter analysis indicate that most *ZmIDHs* can respond to multiple abiotic stresses.

Furthermore, the large variance in the gene structure, phylogenetic tree, conserved protein motifs, protein tertiary structures, and predicted promoter elements for ZmIDHs suggests that individual ZmIDH proteins may be involved in different metabolic pathways to regulate plant growth and development in maize, especially in response to abiotic stress. Most IDHs in higher plants act to regulate the response to abiotic stress, such as low temperature [40] and drought stress [41]. The activity of *NADPH-IDH* can increase by 75% under low-temperature conditions in rice, leading to a direct increase in the content of α-ketoglutarate to trigger the production of proline by stimulating the activity of glutamine synthase [40]. For the characterization of *ZmIDH* genes in maize, *Zm00001d011487*, a *NADP-IDH* gene named *ZmIDH2.4* in this study, was strongly expressed in roots and young embryos and expressed more strongly under salt and drought stresses [24]. Furthermore, when overexpressed, the fertility of the *Zm00001d008244* gene (*ZmIDH1.5* in this study) in *Arabidopsis* will decrease [27]. Additionally, the identification of *ZmIDH1.5* (*Zm00001d008244*) as an interacting protein with the drought-responsive gene *ZmSRO1d* suggests a potential involvement in drought responses [42]. Thus, these IDH genes can respond to stress and changes in development.

In this big genome data era, there has been much public genomic and phenotypic data on maize since the B73 reference genome sequence was released. Whole-genome sequencing has been extensively utilized for candidate gene association analyses for gene family members. Liu et al. identified a natural variant of *ZmDREB* genes from transcriptome data of an association panel and discovered that *ZmDREB2.7* is associated with drought resistance at the seedling stage, a finding that was validated by molecular biological experiments [43]. Mapping for members of a gene family using the SNPs around or located in the candidate gene region from high-throughput genomic or transcriptome data can be highly effective. Here, we utilized publicly available data, including transcriptome data, phenotypic data, and genotype data, to explore the relationships of natural variants of *ZmIDHs* with drought resistance at the seedling stage, using seed survival rate as an indicator. We found more than one significantly associated SNPs around or located in *ZmIDH* genes at the *p* < 0.05 level, except *ZmIDH1.3* and *ZmIDH2.1*. Additionally, two SNPs around the *ZmIDH1.4* significantly are associated with the seed survival rate phenotype at *p* ≤ 0.001 level (Figure 8). Meanwhile, the expression of *ZmIDH1.4* in multiple tissues also showed more sensitive responses to abiotic stress, including drought, PEG, and ABA stress (Figure 7). All these results suggest that these genes may regulate multiple abiotic stresses at different developmental stages. Further work should investigate the detailed mechanisms of action of these important genes.

## 4. Conclusions

In this study, 11 *ZmIDHs* in the maize B73 latest reference genome were identified and classified into two subfamilies according to their coenzymes. By comprehensive analysis, we found that the different types of *ZmIDHs* differed greatly in physicochemical properties, gene structure, promoter cis-elements, conserved motifs, and protein tertiary structure, with great similarity with members of the same type of *ZmIDH* genes. Furthermore, the expression pattern analysis showed that most *ZmIDHs* varied their expression in response to a variety of abiotic stresses, and the responsive tissue and stage varied. Furthermore, the candidate gene association analysis using the public phenotype and sequence data in the association panel with 368 inbred lines showed that natural variation in *ZmIDHs* contributes to drought tolerance in maize seedlings, especially for *ZmIDH1.4*. All these results will provide an important basis for understanding how *ZmIDHs* respond to stress.

## 5. Materials and Methods

### 5.1. Identification of IDH Genes

Using a Hidden Markov Model (HMM), the conserved protein structural domain isocitrate dehydrogenase (PF00180) was obtained from a search of the PFAM database (http://pfam-legacy.xfam.org/ (accessed on 1 September 2022)). The BioMart tool in the Ensembl plants database (https://ensembl.gramene.org/Zeamays/Info/Index (accessed on 1 September 2022)) and the BLASTP tool in the phytozome database (https://phytozome-next.jgi.doe.gov/ (accessed on 1 September 2022)) were used to find the IDH family genes in maize, and the results from the two databases were combined to get all of the IDH family gene IDs and protein sequences in maize. The ExPASy ProtParam (https://web.expasy.org/protparam/ (accessed on 1 September 2022)) was used to determine the length, mass, isoelectric point (pI), and grand average of hydropathicity (GRAVY) of ZmIDH protein sequences. The subcellular localization of ZmIDH protein was predicted using WoLF PSORT II (https://www.genscript.com/wolf-psort.html?src=leftbar (accessed on 1 September 2022)). Using the TBtools tool [44] for bi-directional BLASTP, the homologous *ZmIDH* sequences in rice and sorghum were discovered and used to build the phylogenetic tree by setting the Bootstrap repeat as 1000 times by the neighbor-joining method. The phylogenetic tree was embellished using the online tool evolview (http://www.evolgenius.info/evolview (accessed on 1 September 2022)).

### 5.2. Characterization of Gene Structure, Protein Tertiary Structure, and Motif Patterns

Sequences of promoters, CDS, and amino acid sequences were obtained from the Ensembl plants database (https://ensembl.gramene.org/Zea_mays/Info/Index (accessed on 1 September 2022)). The online tool MEME Suite (https://meme-suite.org/meme/ (accessed on 1 September 2022)) was used to identify the conserved ZmIDH motifs. The PlantCARE online tool (https://bioinformatics.psb.ugent.be/webtools/plantcare/html/ (accessed on 1 September 2022)) was used to predict transcription factor binding sites for ZmIDH. TBtools software was used to construct phylogenetic trees, predict gene structures, and find conserved motifs. The online tool SWISS-MODEL (https://swissmodel.expasy.org/ (accessed on 1 September 2022)) was used to predict the tertiary structure of proteins. The bar and stacked charts were drawn using the online tool ChiPlot (https://www.chiplot.online/#Line-plot (accessed on 1 September 2022)).

### 5.3. Chromosomal Distribution and Gene Duplication of ZmIDH Family Genes

GFF files of maize, rice, and sorghum were downloaded from the phytozome database and analyzed and mapped for colinearity of maize IDH family genes using TBtools software. Maize, rice, and sorghum genomes were compared using One-Step MCScanX-Super Fast, with values calculated for all possible pairs of homologous genes between species. Using the DnaSP6 software, the ratios of nonsynonymous to synonymous substitutions (ka/ks) in paralogous genes were studied [45].

### 5.4. Prediction and Correlation Analysis of IDH-Interacting Proteins

The protein-protein interactions were predicted using the String database (https://cn.string-db.org/ (accessed on 1 September 2022)). And, the online tool omicshare (https://www.omicshare.com/tools/ (accessed on 1 September 2022)) was used to assess the enrichment of GO and KEGG terms. Protein regulatory networks were visualized and embellished with Cytoscape 3.8.2 software.

### 5.5. Expression Analysis of ZmIDH Genes

The RNA-Seq data in various tissues and under drought stress in previous studies were collected in the bar database (http://bar.utoronto.ca/ (accessed on 1 September 2022)) [34,35]. Our unpublished transcriptome data from maize seedlings under ABA and PEG stress were used to extract FPKM data. Furthermore, the expression of *ZmIDH* genes was extracted for further analysis in this study. For tissue-specific analysis of *ZmIDH*, different tissues, including anther, silk, ear leaf, ear, and tassel, were taken from KA105 and B73 inbred lines during flowering. The ChiPlot tool was used to create heat maps.

KA105-inbred seeds were germinated on filter paper dampened with distilled water at 26 °C. The germinated seeds were then moved to a liquid medium with Hoagland, and when the seedlings reached the three-leaf-one stage, they were exposed to abiotic stress. Abiotic stress was applied using three methods: with 200 mM NaCl, 15% PEG6000, and in a liquid medium without nitrogen. Samples were taken at 0, 24, 48, and 72 h after the stress. For transcriptional analysis, leaf and root samples were stressed with 100 m ABA and 10% PEG and then taken at 0, 6, 8, and 12 h after the stress. For each experiment, three biological replicas were used.

The RNAprep Pure Plant Total RNA Extraction Kit was used to extract total RNA from various seedling samples (TIANGEN, China). Using the FastKing RT Kit (with gDNase), extracted RNA was reverse transcribed into first-strand cDNA (TIANGEN, Beijing, China). All *ZmIDH* genes were subjected to RT-qPCR to determine their expression levels. The 2^−ΔΔCT^ method was used to quantify the RT-qPCR data [46]. GraphPad Prism 9.0 was used for statistical analysis and graphing.

### 5.6. Candidate Gene Association Analysis and Haplotype Analysis

The physical locations of the 11 *ZmIDHs* were obtained from the online database MaizeGDB (https://www.maizegdb.org/ (accessed on 1 September 2022)) using the B73 reference genome verb 5.0. All SNPs are located in the promoters and gene regions for the 11 *ZmIDHs* and i-traits in the previous study were obtained from the public website (https://figshare.com/articles/dataset/Genotype_phenotypic_image_data_2017_maize/14429003/1 (accessed on 1 September 2022)) as described previously [29], with a minimum allele frequency (MAF) greater than 0.05, using the MLM model in TASSEL 5.0 software. The significantly associated SNPs were used for haplotype analysis. Phenotypic differences between different haplotypes were calculated using Microsoft Excel 2021. Graphs were edited and integrated using Adobe Illustrator 2021.

## Figures and Tables

**Figure 1 plants-12-03466-f001:**
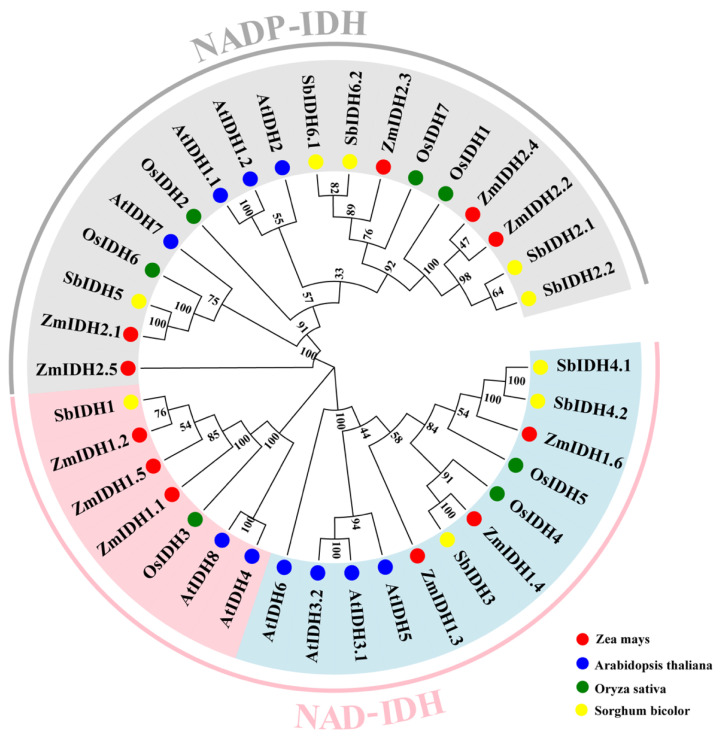
Phylogenetic relationship of IDH proteins in maize, *Arabidopsis*, rice and sorghum Gray is NADP-IDH, pink is catalytic subunit NAD-IDH, and blue is non-catalytic subunit NAD-IDH.

**Figure 2 plants-12-03466-f002:**
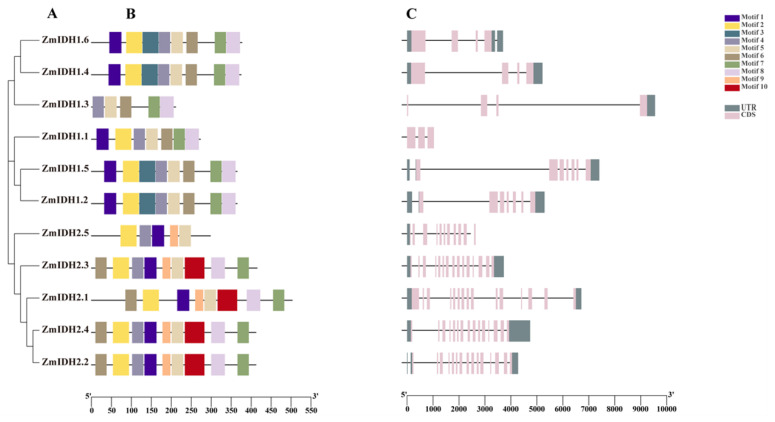
Phylogenetic relationships, conserved protein motifs and gene structures of the 11 ZmIDHs. (**A**): The phylogenetic tree was constructed based on the full-length sequences of maize IDH proteins using MEGA 7. (**B**): Motifs distribution of the IDH proteins. The motifs, numbered 1–10, are displayed in different coloured boxes. (**C**): Exon-intron structures of the *ZmIDH* genes. Pink boxes indicate exons; black lines indicate introns.

**Figure 3 plants-12-03466-f003:**
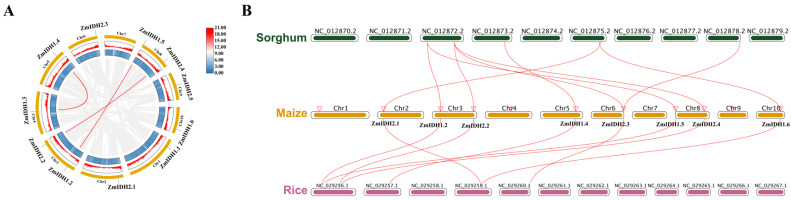
Gene duplication and colinearity analysis of the IDH genes in maize. (**A**). Chromosome distribution and colinearity analysis of the *ZmIDH* gene. The chromosome distribution of the *ZmIDH* gene was mapped using the genome visualization tool of TBtools, and the representative segmental duplication genes connected by lines. Color gradients in chromosomes indicate gene density. (**B**). Colinearity analysis of *Zea mays* with *Oryza sativa* and *Sorghum bicolor.* The red curve represents IDH genes with collinearity.

**Figure 4 plants-12-03466-f004:**
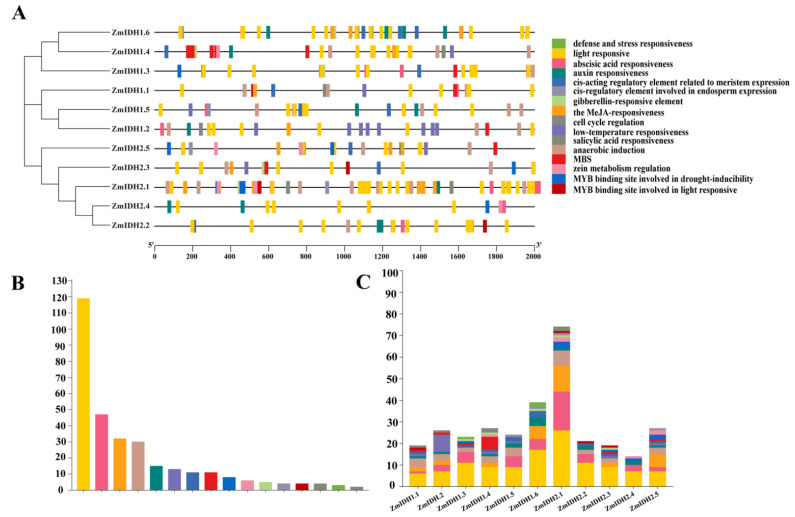
Prediction of *ZmIDH* gene promoter cis-acting elements. The 2000 bp upstream of the start codon was downloaded from the Ensembl plant database. (**A**). The cis-acting elements of 11 *ZmIDH* promoters in maize were analyzed on the PlantCARE website. (**B**). Distribution of the number of cis-regulatory elements in the promoter of *ZmIDH* gene. (**C**). The distribution of various elements in the promoter region of *ZmIDH* gene is indicated by different colors.

**Figure 5 plants-12-03466-f005:**
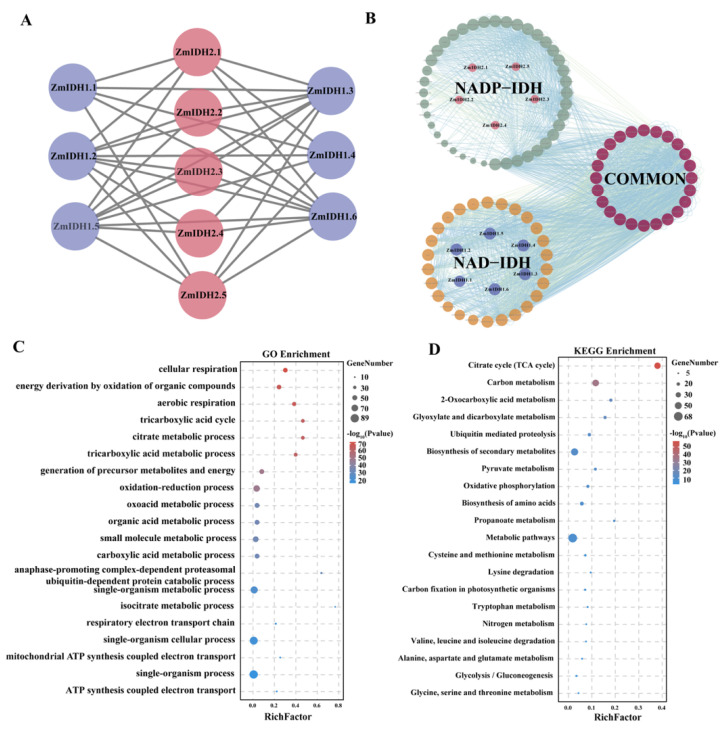
Interactome and GO analysis of ZmIDH proteins. (**A**). Analysis of interactions among 11 ZmIDH proteins. The gray color represents the catalytic subunit NAD-IDH, the purple color represents the non-catalytic subunit NAD-IDH, and the green color represents NADP-IDH. (**B**). Protein interaction network between ZmIDH and other proteins. The green color represents proteins interacting with NADP-IDH, the orange color represents proteins interacting with NAD-IDH, the red color represents proteins interacting with both NAD-IDH and NADP-IDH, the pink color represents NADP-IDH, and the blue color represents NAD-IDH. (**C**). Enrichment analysis of GO terms for interacting proteins. (**D**). Enrichment analysis of interacting proteins using the KEGG database.

**Figure 6 plants-12-03466-f006:**
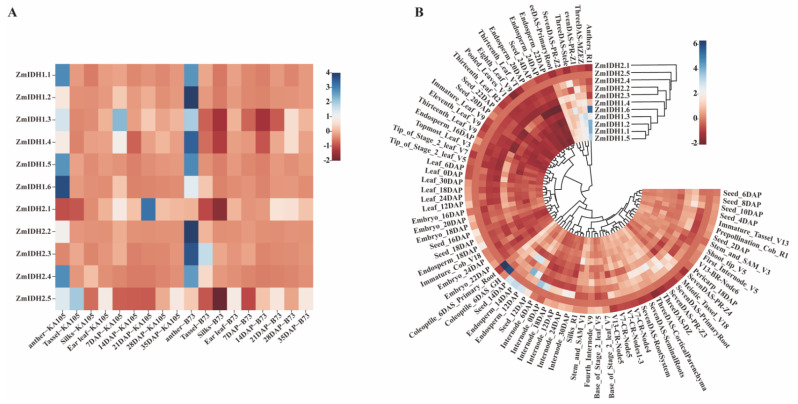
Expression pattern of *ZmIDH* gene in different tissues. (**A**). Expression pattern of *ZmIDH* gene in different tissues in KA105 and B73. Sampling time was at flowering stage and tissues sampled included anther, silks, tassel, ear leaf, seeds at 7, 14, 21, 27, and 35 days after pollination. (**B**). RNA-seq data of *ZmIDH* gene under different tissues were obtained from the bar database (http://bar.utoronto.ca/, accessed on 1 September 2022).

**Figure 7 plants-12-03466-f007:**
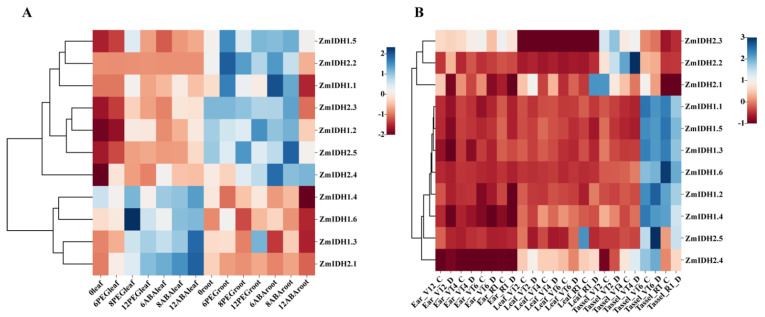
Expression patterns of *ZmIDHs* in response to different abiotic stresses based on RNA-seq data. (**A**). *ZmIDHs* expression data were obtained from our group’s results of leaf and root sampling and sequencing at 0 h, 6 h, 8 h and 12 h after PEG and ABA stresses at the trifoliate stage of maize. (**B**). Expression data of *ZmIDHs* in ear, leaf and tassel at different growth and developmental periods in control and after drought stress were obtained from published data [35].

**Figure 8 plants-12-03466-f008:**
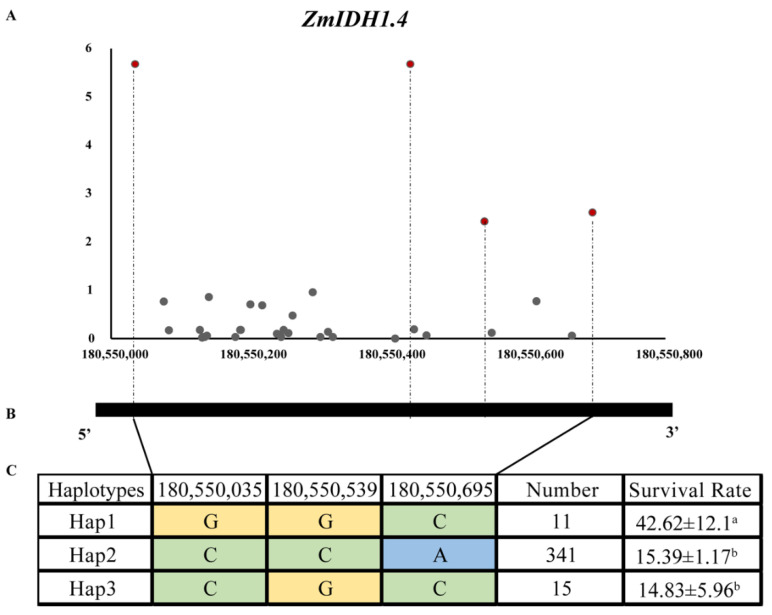
Association analysis and haplotype analysis of *ZmIDH1.4* with seed suirvival rate under drought in maize. (**A**). SNP distribution in *ZmIDH1.4*, red dots indicate significant SNPs associated with drought tolerance. (**B**). Gene structure figure of *ZmIDH1.4*. (**C**). Three haplotypes of *ZmIDH1.4* and their corresponding phenotypic values. Different colors represent different alleles. Superscript lowercase letters represent significant differences of multiple comparisons.

**Table 1 plants-12-03466-t001:** Basic information of IDH gene family in Maize.

Gene Name	Gene ID	Position	Protein Length (aa)	pI	MW	GRAVY	Subcellular Location
*ZmIDH1.1*	Zm00001d028735	Chr1:44718501–44719536	273	6.72	29,983.39	−0.158	Mitochondrion
*ZmIDH1.2*	Zm00001d040438	Chr3:43449762–43455057	365	6.52	39,724.58	−0.091	Mitochondrion
*ZmIDH1.3*	Zm00001d050965	Chr4:134060297–134069849	211	9.21	22,949.69	−0.068	Mitochondrion
*ZmIDH1.4*	Zm00001d017091	Chr5:185084105–185089320	375	8.91	40,361.7	−0.049	Mitochondrion
*ZmIDH1.5*	Zm00001d008244	Chr8:2452376–2459944	365	6.33	39,907.84	−0.109	Mitochondrion
*ZmIDH1.6*	Zm00001d025690	Chr10:126319920–126323617	377	6.72	40,565.61	−0.046	Mitochondrion
*ZmIDH2.1*	Zm00001d003083	Chr2:31519654–31526365	503	8.24	56,084.57	−0.179	Cytoplasmic
*ZmIDH2.2*	Zm00001d044021	Chr3:216967820–216972096	412	6.57	46,097.73	−0.231	Cytoplasmic
*ZmIDH2.3*	Zm00001d039079	Chr6:169685197–169688921	415	6.24	46,223.72	−0.238	Cytoplasmic
*ZmIDH2.4*	Zm00001d011487	Chr8:152072966–152077705	412	6.11	46,196.77	−0.245	Cytoplasmic
*ZmIDH2.5*	Zm00001d046262	Chr9:77157097–77159734	298	8.19	33,719.93	−0.116	Cytoplasmic; Chloroplast

**Table 2 plants-12-03466-t002:** Association analysis of the *ZmIDH* genes with seed survival rate of seedlings under drought in maize.

Gene ID	Gene Name	Polymorphic Number	MLM (*p* ≤ 0.05)	MLM (*p* ≤ 0.01)	MLM (*p* ≤ 0.001)	Max R^2^ (%)
Zm00001d028735	*ZmIDH1.1*	14	3	1	0	1.956
Zm00001d040438	*ZmIDH1.2*	30	1	0	0	1.611
Zm00001d050965	*ZmIDH1.3*	23	0	0	0	1.029
Zm00001d017091	*ZmIDH1.4*	32	4	4	2	6.706
Zm00001d008244	*ZmIDH1.5*	27	4	1	0	2.094
Zm00001d025690	*ZmIDH1.6*	55	3	1	0	2.329
Zm00001d003083	*ZmIDH2.1*	20	0	0	0	1.024
Zm00001d044021	*ZmIDH2.2*	42	1	1	0	2.372
Zm00001d039079	*ZmIDH2.3*	66	10	1	0	2.032
Zm00001d011487	*ZmIDH2.4*	28	1	0	0	1.833
Zm00001d046262	*ZmIDH2.5*	52	1	0	0	1.213

## Data Availability

All data used in this study are available in public databases, as described in the Materials and Methods section.

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
