# Peer review of "Characterization of the Isocitrate Dehydrogenase Gene Family and Their Response to Drought Stress in Maize"

_plants, 2023, doi:10.3390/plants12193466_

Round 1
Reviewer 1 Report
Answering specific question
Is the research design appropriate?
The use of published data is not adequate once the manuscript already has enough data to present. The published data has to be used to support the discussion!
Are the methods adequately described?
As most of the analysis is in silico, an overall improvement of the explanation of the methods would be suggested
Are the results clearly presented?
No. All figures need a deep review, to improve the overall quality. More details are in the manuscript and in the comments. Additionally, the results are mixed with superficial discussions.
Are the conclusions supported by the results?
Must be improved because the discussion also must be improved.
The comments clearly show the low quality of presentation:
The title does not clearly describe the manuscript. It is important to highlight that a great part of the manuscript was in silico analyses or reusing published data. The only experimental procedure was the qPCR analysis. Then, all inferences in the text regarding to IDH genes should be putative, predicted or candidate, unless any of these genes were previously cloned and functionally characterized, but no by this manuscript.
IDH in the title should be named as isocitrate dehydrogenase,
Results:
There are several conclusions, discussions or inferences after each result, which make them weak and superficial. Several examples are highlighted in the text. My general suggestion is to save all of them and organize a deep discussion based on the overall results for each candidate gene.
Figures: All figures require enlargement of font size of all names, captions, and legends. This problem is more severe for figures with several panels. Revise if all panels are necessary. Improve the caption of all figures.
Figure 2: A weird alignment can be noted in the panel 2B. Based on the motif’s alignment, visually, the motif distribution on protein 2.3 is more similar to 2.4 and 2.2 than the 2.1. Please check.
Phrase 129: the phrase has a strong mistake, mixing genes x proteins x groups of genes. Exons and introns are related to genes, not to proteins! …two different… group of genes, not proteins! Please revise the whole phrase.
Figure 3 requires a strong reorganization. My suggestion is to represent the genomes in circles and clearly show the “segmental duplication” in maize as well as the relationship among IDH genes from maize, sorghum and rice. As the data are representations in silico, a better presentation is required. Several questions were raised regarding to these results. The grey lines in panel A do not support the segmental duplication. On the other hand, the excess of grey curves in panel B make confusion.
Line 158: This phrase requires a deep review. “The two catalytically active NAD-IDH genes ZmIDH1.2 and ZmIDH1.5 are segmental duplications, …” This phrase states that three pairs of candidates IDH genes are segmental duplications, without data support. The phrase is written after Fig 3A. Improve the explanation of the collinearity analysis in the panel 3A. Explain also what represent the grey lines linking the pairs of genes. This information is required to clearly explain why the three pairs of IDH genes are considered segmental duplications. So far, the results of collinearity among maize, sorghum and rice are shown in the panel 3B.
Some information that I could extract: the sequence similarity for genes 1.2 and 1.5 can be found in Figs 1 and 2B. For genes 2.2 and 2.4, the panel 3B shows the collinearity, and also the Figures 2A and 2B show sequence similarity. A suggestion in the panel 3B is to add the same color for the genes 2.2 and 2.4, and eliminate all grey lines, which show none information. For genes 1.3 and 1.4, only the fig 1 shows sequence similarity between these candidate genes.
Line 165: The discussion on gene number evolution in the results is very superficial, as found in other parts of the results. Additionally, repetitive DNA in a genome is not enough to support this kind of gene evolution.
Figure 5: Graphs in panels A and B are very confused. Many lines linking colored dots with no relevant information. Similar colors in these panels have no relationship, which make them even more confused. No relevant information can be extracted of these tangles of lines. On the other hand, the names on panels C and D are unreadable.
Figures 6 and 7 have part of your data and part of data already published. Additionally, all these data (names) are unreadable. Thus, my strong suggestion is to use the published data to improve your discussion and show appropriately only your data!
The whole discussion should be reorganized, as the suggestions in the manuscript.

The English should be revised (from moderate to extensive). Several phrases need to be reworded. Some suggestions were added in the manuscript. The references of IDH as genes should be in italics, please check in the whole manuscript.
Author Response
请参阅附件。

Reviewer 2 Report
The authors have made commendable efforts in characterizing the IDH gene family in maize and its response to drought stress. However, the current title lacks clarity. I recommend revising it to reflect the focus on the response of the IDH gene family to drought stress.
Abstract:
Furthermore, the sentence mentioning classification based on coenzymes is unclear; the authors should provide additional context or clarify whether they determined coenzyme activity.
Introduction:
While the introduction sets the stage for the study, it would be beneficial for the authors to expand on previous research related to the IDH gene family. Incorporating a brief overview of similar studies published earlier could provide readers with a broader perspective and highlight the novelty of the current research.
Methods:
To enhance the robustness of the phylogenetic distribution analysis, I suggest employing more advanced methods such as RAxML or MCMC-based approaches. This would provide a clearer and more reliable representation of the phylogenetic relationships within the IDH gene family. Additionally, the absence of bootstrap values in the phylogenetic tree limits its interpretability. Consider providing Figure 3 as supplementary material, as the syntenic relation of a small number of genes may not warrant inclusion in the main document.
Abstract:
The phrasing in the abstract appears unclear and contains a typographical error. Consider revising the sentence to read: "Based on differences in their respective coenzymes, ZmNAD-IDH...". This will enhance clarity and accuracy regarding the classification of ZmNAD-IDH genes based on their coenzymes.
Inadequate support for the claim of "natural variation in ZmIDH1.4 may be associated with drought tolerance":
The assertion regarding the potential link between natural variation in ZmIDH1.4 and drought tolerance requires more substantial evidence. Incorporating additional data or comparative analyses could strengthen this claim and provide more convincing support.
While the study presents valuable insights into the IDH gene family's response to drought stress in maize, there are several areas that need improvement. Addressing the issues highlighted above will undoubtedly enhance the clarity, reliability, and overall impact of the research article.
No specific comment
Author Response
请参阅附件。

Round 2
Reviewer 1 Report
Figure 1: the previous phylogenetic tree, based on gene sequences, better displayed the NADP x NAD candidate genes than the current tree based on proteins. I don´t know why it was changed but I preferred the tree based on genes. Interestingly, the tree changed but the results maintained the same.
Line 132: Physicochemical is not a correct description for these results, which are in silico prediction of gene and protein structures. Please, change this subtitle.
Figure 2: This figure needs a deep reformulation. According to the figure caption, the phylogenetic tree (A) was based on protein sequence, but it is closer to gene organization (B) than to the protein motifs (C). Additionally, the phylogenetic relationship does not match with the protein motifs. See, protein 1.4 and 1.3 were closer in the tree but sharing different motifs compared with 1.6 and 1.4. Similarly, 2.3 and 2.5 are closer in the tree but with very different motifs. Please revise this figure.
Suggestion: the paragraph 136 – 141 should be eliminated, very weak inferences based on intron-exon organization. Use gene sequences to compare IDH genes in figure 1, which were much better in the previous version than using protein sequence. Use protein sequences and motifs only, in figure 2. Also revise the comparisons between motif organization and position in the phylogenetic tree.

Many corrections made by the authors have misspelling. This these kinds of mistakes are not acceptable, and it is not a reviewer job to correct English grammar or misspelling. Thus, a deep professional English review in mandatory.
Only to highlight a few examples in the beginning of the manuscript:
Line 12: “IDH also participants in plant growth…”, which should be participates
Line 84: “the increasement of”
Line 90: were considered to work plant…, which should be to work on…
Author Response
请参阅附件。

Reviewer 2 Report
The revised manuscript appears to be well-aligned with the feedback provided. The authors have diligently addressed all of the concerns raised. In light of these revisions, I recommend accepting the manuscript in its current form.
no comment
Author Response
Thank you for serving as reviewers to consider our manuscript for publication in Plants. We would like to thank you for the time and effort to review our manuscript. We invited native speakers to improve the language and some spelling errors have been corrected. Thank you very much!